

# Reanalyzing Head et al. (2015): investigating the robustness of widespread *p*-hacking

Chris H.J. Hartgerink

Department of Methodology and Statistics, Tilburg University, Tilburg, The Netherlands

## ABSTRACT

*Head et al. (2015)* provided a large collection of *p*-values that, from their perspective, indicates widespread statistical significance seeking (i.e., *p*-hacking). This paper inspects this result for robustness. Theoretically, the *p*-value distribution should be a smooth, decreasing function, but the distribution of reported *p*-values shows systematically more reported *p*-values for .01, .02, .03, .04, and .05 than *p*-values reported to three decimal places, due to apparent tendencies to round *p*-values to two decimal places. *Head et al. (2015)* correctly argue that an aggregate *p*-value distribution could show a bump below .05 when left-skew *p*-hacking occurs frequently. Moreover, the elimination of $p = .045$ and $p = .05$, as done in the original paper, is debatable. Given that eliminating $p = .045$ is a result of the need for symmetric bins and systematically more *p*-values are reported to two decimal places, I did not exclude $p = .045$ and $p = .05$. I conducted Fisher's method $.04 < p < .05$ and reanalyzed the data by adjusting the bin selection to $.03875 < p \leq .04$ versus $.04875 < p \leq .05$. Results of the reanalysis indicate that no evidence for left-skew *p*-hacking remains when we look at the entire range between $.04 < p < .05$ or when we inspect the second-decimal. Taking into account reporting tendencies when selecting the bins to compare is especially important because this dataset does not allow for the recalculation of the *p*-values. Moreover, inspecting the bins that include two-decimal reported *p*-values potentially increases sensitivity if strategic rounding down of *p*-values as a form of *p*-hacking is widespread. Given the far-reaching implications of supposed widespread *p*-hacking throughout the sciences *Head et al. (2015)*, it is important that these findings are robust to data analysis choices if the conclusion is to be considered unequivocal. Although no evidence of widespread left-skew *p*-hacking is found in this reanalysis, this does not mean that there is no *p*-hacking at all. These results nuance the conclusion by *Head et al. (2015)*, indicating that the results are not robust and that the evidence for widespread left-skew *p*-hacking is ambiguous at best.

Corresponding author
Chris H.J. Hartgerink,
c.h.j.hartgerink@tilburguniversity.edu,
chjh@protonmail.com

## INTRODUCTION

*Head et al. (2015)* provided a large collection of *p*-values that, from their perspective, indicates widespread statistical significance seeking (i.e., *p*-hacking) throughout the sciences. This result has been questioned from an epistemological perspective because analyzing all reported *p*-values in research articles answers the supposedly inappropriate question of evidential value across all results (*Simonsohn, Simmons & Nelson, 2015*).

Adjacent to epistemological concerns, the robustness of widespread $p$-hacking in these data can be questioned due to the large variation in a priori choices with regards to data analysis. *Head et al. (2015)* had to make several decisions with respect to the data analysis, which might have affected the results. In this paper I evaluate the data analysis approach with which *Head et al. (2015)* found widespread $p$-hacking and propose that this effect is not robust to several justifiable changes. The underlying models for their findings have been discussed in several preprints (e.g., *Bishop & Thompson, 2015*; *Holman, 2015*) and publications (e.g., *Simonsohn, Simmons & Nelson, 2015*; *Bruns & Ioannidis, 2016*), but the data have not extensively been reanalyzed for robustness.

The $p$-value distribution of a set of true- and null results without $p$-hacking should be a mixture distribution of only the uniform $p$-value distribution under the null hypothesis $H_0$ and right-skew $p$-value distributions under the alternative hypothesis $H_1$. $P$-hacking behaviors affect the distribution of statistically significant $p$-values, potentially resulting in left-skew below .05 (i.e., a bump), but not necessarily so (*Hartgerink et al., 2016*; *Lakens, 2014*; *Bishop & Thompson, 2016*). An example of a questionable behavior that can result in left-skew is optional stopping (i.e., data peeking) if the null hypothesis is true (*Lakens, 2014*).

Consequently, *Head et al. (2015)* correctly argue that an aggregate $p$-value distribution could show a bump below .05 when left-skew $p$-hacking occurs frequently. Questionable behaviors that result in seeking statistically significant results, such as (but not limited to) the aforementioned optional stopping under $H_0$, could result in a bump below .05. Hence, a systematic bump below .05 (i.e., not due to sampling error) is a sufficient condition for the presence of specific forms of $p$-hacking. However, this bump below .05 is not a necessary condition, because other types of $p$-hacking can still occur without a bump below .05 presenting itself (*Hartgerink et al., 2016*; *Lakens, 2014*; *Bishop & Thompson, 2016*). For example, one might use optional stopping when there is a true effect or conduct multiple analyses, but only report that statistical test which yielded the smallest $p$-value. Therefore, if no bump of statistically significant $p$-values is found, this does not exclude that $p$-hacking occurs at a large scale.

In the current paper, the conclusion from *Head et al. (2015)* is inspected for robustness. Their conclusion is that the data fullfill the sufficient condition for $p$-hacking (i.e., show a systematic bump below .05), hence, provides evidence for the presence of specific forms of $p$-hacking. The robustness of this conclusion is inspected in three steps: (i) explaining the data and data analysis strategies (original and reanalysis), (ii) reevaluating the evidence for a bump below .05 (i.e., the sufficient condition) based on the reanalysis, and (iii) discussing whether this means that there is no widespread $p$-hacking in the literature.

## DATA AND METHODS

In the original paper, over two million reported $p$-values were mined from the Open Access subset of PubMed central. PubMed central indexes the biomedical and life sciences and permits bulk downloading of full-text Open Access articles (https://www.ncbi.nlm.nih.gov/pmc/tools/openftlist/). By text-mining these full-text articles for $p$-values, *Head et al. (2015)* extracted more than two million $p$-values in total. Their text-mining

procedure extracted all reported $p$-values, including those that were reported without an accompanying test statistic. For example, the $p$-value from the result $t(59) = 1.75, p > .05$ was included, but also a lone $p < .05$. Subsequently, *Head et al. (2015)* analyzed a subset of statistically significant $p$-values (assuming $\alpha = .05$) that were exactly reported (e.g., $p = .043$; the same subset is analyzed in this paper).

*Head et al. (2015)* their data analysis approach focused on comparing frequencies in the last and penultimate bins from .05 at a binwidth of .005 (i.e., $.04 < p < .045$ versus $.045 < p < .05$). Based on the tenet that a sufficient condition for $p$-hacking is a systematic bump of $p$-values below .05 (*Simonsohn, Nelson & Simmons, 2014*), sufficient evidence for $p$-hacking is present if the last bin has a significantly higher frequency than the penultimate bin in a binomial test. Applying the binomial test to two frequency bins has previously been used in publication bias research (Caliper test; *Gerber et al., 2010*; *Kühberger, Fritz & Scherndl, 2014*), applied here specifically to test for $p$-hacking behaviors that result in a bump below .05. The binwidth of .005 and the bins $.04 < p < .045$ and $.045 < p < .05$ were chosen by *Head et al. (2015)* because they expected the signal of this form of $p$-hacking to be strongest in this part of the distribution (regions of the $p$-value distribution closer to zero are more likely to contain evidence of true effects than regions close to .05). They excluded $p = .05$ "because [they] suspect[ed] that many authors do not regard $p = 0.05$ as significant" (p. 4).

Figure 1 shows the selection of $p$-values in *Head et al. (2015)* in two ways: (1) in green, which shows the results as analysed by Head et al. (i.e., $.04 < p < .045$ versus $.045 < p < .05$), and (2) in grey, which shows the entire distribution of significant $p$-values (assuming $\alpha = .05$) available to Head et al. after eliminating $p = .045$ and $p = .05$ (depicted by the black bins). The height of the two green bins (i.e., the sum of the grey bins in the same range) show a bump below .05, which indicates $p$-hacking. The grey histogram in Fig. 1 shows a more fine-grained depiction of the $p$-value distribution and does not clearly show a bump below .05, because it is dependent on which bins are compared. However, the grey histogram clearly indicates that results around the second decimal tend to be reported more frequently when $p \geq .01$.

Theoretically, the $p$-value distribution should be a smooth, decreasing function, but the grey distribution shows systematically more reported $p$-values for .01, .02, .03, .04 (and .05 when the black histogram is included). As such, there seems to be a tendency to report $p$-values to two decimal places, instead of three. For example, $p = .041$ might be correctly rounded down to $p = .04$ or $p = .046$ rounded up to $p = .05$. A potential post-hoc explanation is that three decimal reporting of $p$-values is a relatively recent standard, if a standard at all. For example, it has only been prescribed since 2010 in psychology (*APA, 2010*), where it previously prescribed two decimal reporting (*APA, 1983*; *APA, 2001*). Given the results, it seems reasonable to assume that other fields might also report to two decimal places instead of three, most of the time.

Moreover, the data analysis approach used by *Head et al. (2015)* eliminates $p = .045$ for symmetry of the compared bins and $p = .05$ based on a potentially invalid assumption of when researchers regard results as statistically significant. $P = .045$ is not included in the selected bins ($.04 < p < .045$ versus $.045 < p < .05$), while this could affect the results.
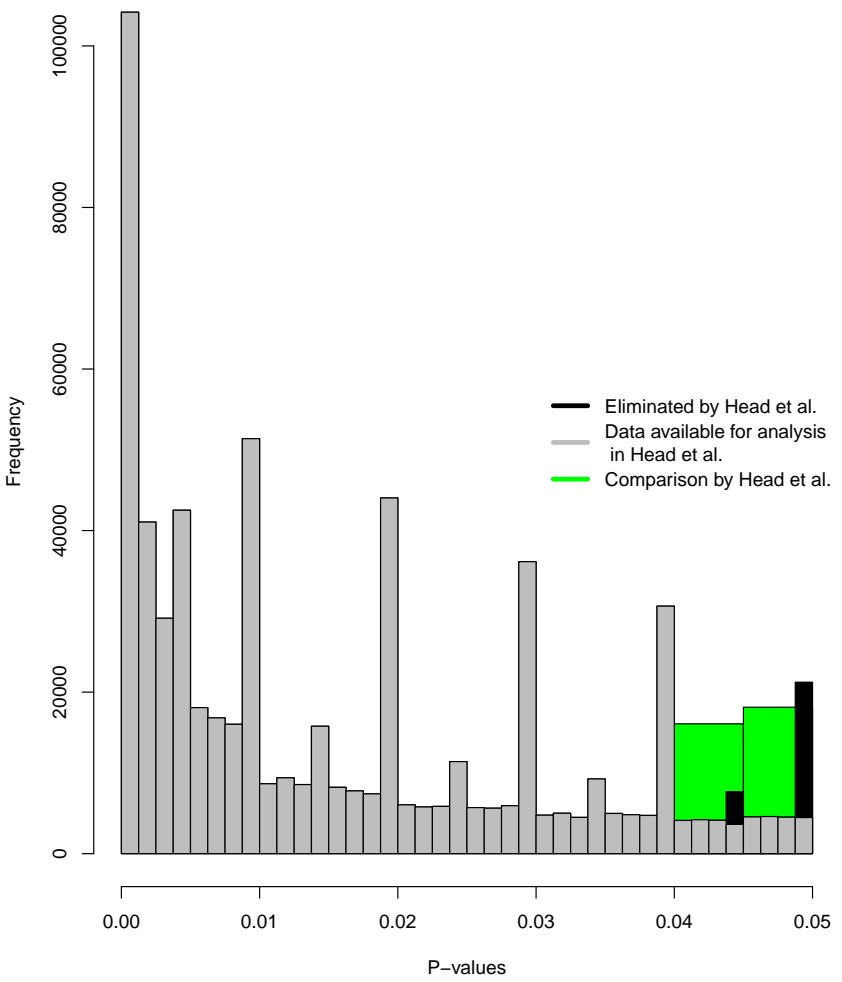

**Figure 1** **Histograms of *p*-values as selected in Head et al. (in green; .04 < *p* < .045 versus .045 < *p* < .05), the significant *p*-value distribution as selected in Head et al. (in grey; 0 < *p* ≤ .00125, .00125 < *p* ≤ .0025, ..., .0475 < *p* ≤ .04875, .04875 < *p* < .05, binwidth = .00125).** The green and grey histograms exclude *p* = .045 and *p* = .05; the black histogram shows the frequencies of results that are omitted because of this (.04375 < *p* ≤ .045 and .04875 < *p* ≤ .05, binwidth = .00125).

If *p* = .045 is included, no evidence of a bump below .05 is found (the left black bin in Fig. 1 is then included; frequency .04 < *p* ≤ .045 = 20,114 versus .045 < *p* < .05 = 18,132). However, the bins are subsequently asymmetrical and require a different analysis. To this end, I supplement the Caliper tests with Fisher's method (*Fisher, 1925*; *Mosteller & Fisher, 1948*) based on the same range analyzed by *Head et al. (2015)*. This analysis includes .04 < *p* < .05 (i.e., it does not exclude *p* = .045 as in the binned Caliper test). Fisher's method tests for a deviation from uniformity and was computed as

$$\chi^2_{2k} = -2 \sum_{i=1}^{k} ln\left(\frac{p_i - .04}{.01}\right) \quad (1)$$

where $p_i$ are the *p*-values between .04 < *p* < .05 Effectively, Eq. (1) tests for a bump between .04 and .05 (i.e., the transformation ensures that the transformed *p*-values range from 0–1 and that Fisher's method inspects left-skew instead of right-skew). *P* = .05 was consistently

excluded by *Head et al. (2015)* because they assumed researchers did not interpret this as statistically significant. However, researchers interpret $p = .05$ as statistically significant more frequently than they thought: 94% of 236 cases investigated by *Nuijten et al. (2015)* interpreted $p = .05$ as statistically significant, indicating this assumption might not be valid.

Given that systematically more $p$-values are reported to two decimal places and the adjustments described in the previous paragraph, I did not exclude $p = .045$ and $p = .05$ and I adjusted the bin selection to $.03875 < p \le .04$ versus $.04875 < p \le .05$. Visually, the newly selected data are the grey and black bins from Fig. 1 combined, where the rightmost black bin (i.e., $.04875 < p \le .05$) is compared with the large grey bin at .04 (i.e., $.03875 < p \le .04$). The bins $.03875 < p \le .04$ and $.04875 < p \le .05$ were selected to take into account that $p$-values are typically rounded (both up and down) in the observed data. Moreover, if incorrect or excessive rounding-down of $p$-values occurs strategically (e.g., $p = .054$ reported as $p = .05$; *Vermeulen et al., 2015*), this can be considered $p$-hacking. If $p = .05$ is excluded from the analyses, these types of $p$-hacking behaviors are eliminated from the analyses, potentially decreasing the sensitivity of the test for a bump.

The reanalysis approach for the bins $.03875 < p \le .04$ and $.04875 < p \le .05$ is similar to *Head et al. (2015)* and applies the Caliper test to detect a bump below .05, with the addition of Bayesian Caliper tests. The Caliper test investigates whether the bins are equally distributed or that the penultimate bin (i.e., $.03875 < p \le .04$) contains more results than the ultimate bin (i.e., $.04875 < p \le .05$; $H_0 : Proportion \le .5$). Sensitivity analyses were also conducted, altering the binwidth from .00125 to .005 and .01. Moreover, the analyses were conducted for both the $p$-values extracted from the abstracts- and the results sections separately.

The results from the Bayesian Caliper test and the traditional, frequentist Caliper test give results with different interpretations. The $p$-value of the Caliper test gives the probability of more extreme results if the null hypothesis is true, but does not quantify the probability of the null- and alternative hypothesis. The added value of the Bayes Factor ($BF$) is that it does quantify the probabilities of the hypotheses in the model and creates a ratio, either as $BF_{10}$, the alternative hypothesis versus the null hypothesis, or vice versa, $BF_{01}$. A $BF$ of 1 indicates that both hypotheses are equally probable, given the data. All Bayesian proportion tests were conducted with highly uncertain priors ($r = 1$, 'ultrawide' prior) using the 'BayesFactor' package (*Morey & Rouder, 2015*). In this specific instance, $BF_{10}$ is computed and values >1 can be interpreted, for our purposes, as: the data are more likely under $p$-hacking that results in a bump below .05 (i.e., left-skew $p$-hacking) than under no left-skew $p$-hacking. $BF_{10}$ values <1 indicate that the data are more likely under no left-skew $p$-hacking than under left-skew $p$-hacking. The further removed from 1, the more evidence in the direction of either hypothesis is available.

## REANALYSIS RESULTS

Results of Fisher's method for all $p$-values between $.04 < p < .05$ and does not exclude $p = .045$ fails to find evidence for a bump below .05, $\chi^2(76492) = 70328.86, p > .999$. Additionally, no evidence for a bump below .05 remains when I focus on the more
**Table 1 Results of the reanalysis across various binwidths (i.e., .00125, .005, .01) and different sections of the paper.**

| | | Abstracts | Results |
|---|---|---|---|
| Binwidth = .00125 | $.03875 < p \le .04$ | 4,597 | 26,047 |
| | $.04875 < p \le .05$ | 2,565 | 18,664 |
| | *Proportion* | 0.358 | 0.417 |
| | *p* | >.999 | >.999 |
| | $BF_{10}$ | <.001 | <.001 |
| Binwidth = .005 | $.035 < p \le .04$ | 6,641 | 38,537 |
| | $.045 < p \le .05$ | 4,485 | 30,406 |
| | *Proportion* | 0.403 | 0.441 |
| | *p* | >.999 | >.999 |
| | $BF_{10}$ | <.001 | <.001 |
| Binwidth = .01 | $.03 < p \le .04$ | 9,885 | 58,809 |
| | $.04 < p \le .05$ | 7,250 | 47,755 |
| | *Proportion* | 0.423 | 0.448 |
| | *p* | >.999 | >.999 |
| | $BF_{10}$ | <.001 | <.001 |

frequently reported second-decimal bins, which could include *p*-hacking behaviors such as incorrect or excessive rounding down to $p = .05$. Reanalyses showed no evidence for left-skew *p*-hacking, $Proportion = .417, p > .999, BF_{10} < .001$ for the Results sections and $Proportion = .358, p > .999, BF_{10} < .001$ for the Abstract sections. Table 1 summarizes these results for alternate binwidths (.00125, .005, and .01) and shows results are consistent across different binwidths. Separated per discipline, no binomial test for left-skew *p*-hacking is statistically significant in either the Results- or Abstract sections (see the Supplemental Information 1). This indicates that the evidence for *p*-hacking that results in a bump below .05, as presented by *Head et al. (2015)*, seems to not be robust to minor changes in the analysis such as including $p = .045$ by evaluating $.04 < p < .05$ continuously instead of binning, or when taking into account the observed tendency to round *p*-values to two decimal places during the bin selection.

## DISCUSSION

*Head et al. (2015)* collected *p*-values from full-text articles and analyzed these for *p*-hacking, concluding that "*p*-hacking is widespread throughout science" (see abstract; *Head et al., 2015*). Given the implications of such a finding, I inspected whether evidence for widespread *p*-hacking was robust to some substantively justified changes in the data selection. A minor adjustment from comparing bins to continuously evaluating $.04 < p < .05$, the latter not excluding .045, already indicated this finding seems to not be robust. Additionally, after altering the bins inspected due to the observation that systematically more *p*-values are reported to the second decimal and including $p = .05$ in the analyses, the results indicate that evidence for widespread *p*-hacking, as presented by *Head et al. (2015)* is not robust to these substantive changes in the analysis. Moreover, the frequency of $p = .05$

is directly affected by *p*-hacking, when rounding-down of *p*-values is done strategically. The conclusion drawn by *Head et al. (2015)* might still be correct, but the data do not undisputably show so. Moreover, even if there is no *p*-hacking that results in a bump of *p*-values below .05, other forms of *p*-hacking that do not cause such a bump can still be present and prevalent (*Hartgerink et al., 2016*; *Lakens, 2014*; *Bishop & Thompson, 2016*).

Second-decimal reporting tendencies of *p*-values should be taken into consideration when selecting bins for inspection because this dataset does not allow for the elimination of such reporting tendencies. Its substantive consequences are clearly depicted in the results of the reanalysis and Fig. 1 illustrates how the theoretical properties of *p*-value distributions do not hold for the reported p-value distribution. Previous research has indicated that when the recalculated *p*-value distribution is inspected, the theoretically expected smooth distribution re-emerges even when the reported *p*-value distribution shows reporting tendencies (*Hartgerink et al., 2016*; *Krawczyk, 2015*). Given that the text-mining procedure implemented by *Head et al. (2015)* does not allow for recalculation of *p*-values, the effect of reporting tendencies needs to mitigated by altering the data analysis approach.

Even after mitigating the effect of reporting tendencies, these analyses were all conducted on a set of aggregated *p*-values, which can either detect *p*-hacking that results in a bump of *p*-values below .05 if it is widespread, but not prove that no *p*-hacking is going on in any of the individual papers. Firstly, there is the risk of an ecological fallacy. These analyses take place at the aggregate level, but there might still be research papers that show a bump below .05 at the paper level. Secondly, some forms of *p*-hacking also result in right-skew, which is not picked up in these analyses and is difficult to detect in a set of heterogeneous results (attempted in *Hartgerink et al., 2016*). As such, if any detection of *p*-hacking is attempted, this should be done at the paper level and after careful scrutiny of which results are included (*Simonsohn, Simmons & Nelson, 2015*; *Bishop & Thompson, 2016*).

## LIMITATIONS AND CONCLUSION

In this reanalysis two limitations remain with respect to the data analysis. First, selecting the bins just below .04 and .05 results in selecting non-adjacent bins. Hence, the test might be less sensitive to detect a bump below .05. In light of this limitation I ran the original analysis from *Head et al. (2015)*, but included the second decimal (i.e., $.04 \leq p < .045$ versus $.045 < p \leq .05$). This analysis also yielded no evidence for a bump of *p*-values below .05, $Proportion = .431, p > .999, BF_{10} < .001$. Second, the selection of only exactly reported *p*-values might have distorted the *p*-value distribution due to reporting tendencies in rounding. For example, a researcher with a *p*-value of .047 might be more likely to report $p < .05$ than a researcher with a *p*-value of .037 reporting $p < .04$. Given that these analyses exclude all values reported as $p < X$, this could have affected the results. There is some indication that this tendency to round up is relatively stronger around .05 than around .04 (a factor of 1.25 approximately based on the original Fig. 5; *Krawczyk, 2015*), which might result in an underrepresentation of *p*-values around .05.

Given the implications of the findings by *Head et al. (2015)*, it is important that these findings are robust to choices that can vary. Moreover, the absence of a bump below .05 seems to be stronger than its presence throughout the literature: a reanalysis of a previous

paper, which found evidence for a bump below .05 (*Masicampo & Lalande, 2012*), yielded no evidence for a bump below .05 (*Lakens, 2014*); two new datasets also did not reveal a bump below .05 (e.g., *Hartgerink et al., 2016*; *Vermeulen et al., 2015*). Consequently, findings that claim there is a bump below .05 need to be robust. In this paper, I explained why a different data analysis approach to the data of *Head et al. (2015)* can be justified and as a result no evidence of widespread *p*-hacking that results in a bump of *p*-values below .05 is found. Although this does not mean that no *p*-hacking occurs at all, the conclusion by *Head et al. (2015)* should not be taken at face value considering that the results are not robust to (minor) choices in the data analysis approach. As such, the evidence for widespread left-skew *p*-hacking is ambiguous at best.

## ACKNOWLEDGEMENTS

Joost de Winter, Marcel van Assen, Robbie van Aert, Michèle Nuijten, and Jelte Wicherts provided fruitful discussion or feedback on the ideas presented in this paper. The end result is the author's sole responsibility.

### Funding
The author received no funding for this work.

### Competing Interests
The author declares there are no competing interests.

### Author Contributions
- Chris H.J. Hartgerink conceived and designed the experiments, performed the experiments, analyzed the data, contributed reagents/materials/analysis tools, wrote the paper, prepared figures and/or tables.

### Data Availability
All supporting files for this article are archived at Zenodo, https://doi.org/10.5281/zenodo.259624.

### Supplemental Information
Supplemental information for this article can be found online at http://dx.doi.org/10.7717/peerj.3068#supplemental-information.

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
